# Physiological and Transcriptomic Analyses of the Effects of Exogenous Lauric Acid on Drought Resistance in Peach (*Prunus persica* (L.) Batsch)

**DOI:** 10.3390/plants12071492

**Published:** 2023-03-29

**Authors:** Binbin Zhang, Hao Du, Sankui Yang, Xuelian Wu, Wenxin Liu, Jian Guo, Yuansong Xiao, Futian Peng

**Affiliations:** College of Horticulture Science and Engineering, Shandong Agricultural University, Taian 271018, China

**Keywords:** *Prunus persica* (L.) Batsch, lauric acid, drought stress, physiological indicators, transcriptome

## Abstract

Peach (*Prunus persica* (L.) Batsch) is a fruit tree of economic and nutritional importance, but it is very sensitive to drought stress, which affects its growth to a great extent. Lauric acid (LA) is a fatty acid produced in plants and associated with the response to abiotic stress, but the underlying mechanism remains unclear. In this study, physiological analysis showed that 50 ppm LA pretreatment under drought stress could alleviate the growth of peach seedlings. LA inhibits the degradation of photosynthetic pigments and the closing of pores under drought stress, increasing the photosynthetic rate. LA also reduces the content of O_2_^−^, H_2_O_2_, and MDA under drought stress; our results were confirmed by Evans Blue, nitroblue tetrazolium (NBT), and DAB(3,3-diaminobenzidine) staining experiments. It may be that, by directly removing reactive oxygen species (ROS) and improving enzyme activity, i.e., catalase (CAT) activity, peroxidase (POD) activity, superoxide dismutase (SOD) activity, and ascorbate peroxidase (APX) activity, the damage caused by reactive oxygen species to peach seedlings is reduced. Peach seedlings treated with LA showed a significant increase in osmoregulatory substances compared with those subjected to drought stress, thereby regulating osmoregulatory balance and reducing damage. RNA-Seq analysis identified 1876 DEGs (differentially expressed genes) in untreated and LA-pretreated plants under drought stress. In-depth analysis of these DEGs showed that, under drought stress, LA regulates the expression of genes related to plant–pathogen interaction, phenylpropanoid biosynthesis, the MAPK signaling pathway, cyanoamino acid metabolism, and sesquiterpenoid and triterpenoid biosynthesis. In addition, LA may activate the Ca^2+^ signaling pathway by increasing the expressions of CNGC, CAM/CML, and CPDK family genes, thereby improving the drought resistance of peaches. In summary, via physiological and transcriptome analyses, the mechanism of action of LA in drought resistance has been revealed. Our research results provide new insights into the molecular regulatory mechanism of the LA-mediated drought resistance of peach trees.

## 1. Introduction

In recent years, due to the increase in greenhouse gases around the world, global drought damage and water deprivation are rising, particularly in extremely arid and semi-arid regions [1,2,3,4]. Drought inhibits plants in various stages of normal growth and development, not only affecting the respiration and light interface effect, but also affecting osmotic regulation, protein interfaces, and material transportation, as well as some physiological processes, thus seriously influencing the survival of crops and their growth and yield [5,6,7]. Therefore, in order to promote plant growth, it is critical to investigate effective drought tolerance solutions [8]. It has been demonstrated that using biological modulators or organic acids to enhance plant growth and drought resistance is a successful and environmentally advantageous approach [9,10].

Many studies have confirmed that abiotic stress in plants is alleviated by the addition of plant growth regulators or organic acids, such as strigolactone, dopamine, salicylic acid (SA), jasmonic acid (JA), methyl jasmonate (MeJA), citric acid, and acetic acid, which can regulate the resistance of plants to abiotic stress [11,12,13,14]. For instance, by encouraging the production of JA and its signaling pathway in plants, exogenous acetic acid can enhance drought resistance in a range of plant species, including rice, maize, rapeseed, and wheat [9]. Recent research has demonstrated that acetic acid-induced modifications in plant physiology and metabolic composition increase the resistance of willow to drought [15]. Previous research has demonstrated that treating *Arabidopsis* with nicotinic acid (NA) increases the plant’s tolerance to drought stress and encourages development [16]. In recent years, studies have shown that β-amino butyric acid (BABA) can improve the drought resistance of rice, cowpea, cherry, and other plants [17,18,19].

Fatty acids are a group of significant physiological chemicals that play a role in the energy conversion, membrane structure, and many of the signaling pathways of cells [20]. The medium-chain fatty acid lauric acid (C12) is more active than caproic acid (C6), caprylic acid (C8), and capric acid (C10) [21]. It was discovered to be the primary antibacterial and antiviral component in human breast milk [22]. As a result, lauric acid (LA) research primarily focuses on its antibacterial and bactericidal properties [23,24,25]. It has been demonstrated that a medium treated with LA may enhance the development of zygotic coconut (*Cocos nucifera* L.) embryos, while also greatly increasing plant growth. A radioactive LA metabolism experiment demonstrated that LA serves as a source of carbon for long-chain fatty acids, and that its principal byproduct is the phospholipid that comprises new cell membranes [26]. LA improved zygotic Syagrus coronata embryos during in vitro growth. Additionally, it was shown that plants treated with LA at the pre-acclimation stage had improved drought resistance via the controlling of stomatal closure, the preservation of leaf water content, and the maintenance of photosynthesis [27]. Through a study on the effects of drought stress on the growth and metabolism of loquat, it was found that among more than 100 leaf metabolites, 9 contributed to the metabolism of loquat under drought conditions, including LA [28].

In recent years, the peach (*Prunus persica*) industry has grown in the global economic market, and peach offers high economic, nutritional, and medicinal value [29]. However, drought seriously threatens peach yield and fruit quality during peach growth and development [30]. Therefore, the drought resistance mechanism of peach is being paid increasing attention. Exogenous LA has been applied to peaches in response to drought stress in agricultural production. So far, no studies have comprehensively explored the physiological and molecular mechanisms by which LA mediates peach drought resistance, and our understanding is still limited.

In this study, peach seedlings were subjected to drought stress, and LA was added at the same time to explore its alleviating effect. Additionally, a comparative transcriptome study was carried out on peaches that had been treated with LA versus those that had not, in order to further explore the molecular basis of LA-induced drought tolerance in peach seedlings. Our research should provide information about how the molecular control of peach drought resistance is mediated by LA, and should aid in the search for more potent ways to reduce the negative impacts of drought stress on peach output.

## 2. Results

### 2.1. Effects of LA on Growth and Membrane Permeability of P. persica under Drought

Figure 1A,B show that peach seedling development was considerably hampered by drought stress. Under drought conditions, peach seedlings’ fresh weight, dry weight, and root activity declined by 38.55%, 55.37%, and 28.02%, respectively, in comparison to CK. The application of exogenous LA at a specific dosage during drought treatment was able to stimulate seedling growth, which increased by 21.99%, 23.94%, and 17.40% in comparison to drought treatment stages, respectively. The findings demonstrate that exogenous LA might reduce the impact of drought treatment on peach seedlings. Drought stress significantly decreased the leaf relative water content of peach seedlings. However, after pretreatment with 50 ppm LA, the leaf water status was maintained, and the relative water content of these leaves was higher than that of the control without LA (Figure 1C). Relative electrical leakage is an important physiological index reflecting the damage degree of plant leaves under stress, and the stability of the cell membrane. Here, the REL of the leaves of peach seedlings increased significantly under drought conditions, reaching 35.59%. Compared with plants without LA, the REL (relative electrolyte leakage) of plants treated with exogenous LA under drought stress conditions was lower, and the REL under drought stress was 15.28% with exogenous LA (Figure 1D).

### 2.2. Characteristics of Photosynthetic Parameters and Photosynthetic Pigments of Leaves

As shown in Figure 2, drought stress significantly inhibited the net photosynthetic rate (Pn) of peach leaves. However, the exogenous application of 50 ppm LA increased the photosynthetic capacity of the plants (Figure 2A). The change trend of the stomatal conductance (Gs) value was basically consistent with Pn, showing that Gs decreased significantly under drought stress, while exogenous LA could maintain this at a relatively high level (Figure 2B). Similarly, the intercellular carbon dioxide concentration (Ci) and transpiration rate (Tr) were significantly decreased under drought stress, and exogenous LA alleviated the effects of drought on peach seedlings (Figure 2C,D). As shown in Figure 2, the contents of chlorophyll a (Chl a), chlorophyll b (Chl b), chlorophyll a + b (Chl a + b), and carotenoids (Car) in peach seedlings under drought stress were significantly lower than those under CK (*p* < 0.05), which values decreased by 22.37%, 16.22%, 20.85%, and 30.16% compared with CK treatment, respectively. These results indicate that drought stress could inhibit the accumulation of photosynthetic pigments in peach seedlings. When LA was applied in the drought treatment, the contents of Chl a, Chl b, Chl a + b, and Car were increased by 10.85%, 8.83%, 8.35%, and 20.55%, respectively, indicating that LA significantly alleviated the inhibition of photosynthetic pigment synthesis or degradation in peach seedlings under drought stress. Thus, the photosynthesis of peach seedlings was effectively improved. 

### 2.3. The Effects of LA on the Osmotic Regulation Substances of P. persica under Drought

The content of osmotic regulatory substances in plants reflects their stress resistance to a certain extent. Under drought stress, peach seedlings had considerably higher amounts of soluble sugar, soluble protein, proline, and free amino acids compared to normal circumstances (Figure 3). After drought treatment, the contents of soluble sugar, soluble protein, proline, and free amino acids in peach seedlings increased by 22.43%, 27.03%, 16.36%, and 52.39%, respectively, compared with CK. Compared with drought treatment, the contents of soluble sugar, soluble protein, proline, and free amino acids increased by 22.02%, 13.82%, 21.95%, and 9.46%, respectively, after applying 50 ppm LA. 

### 2.4. Effects of Cell Damage and ROS (Reactive Oxygen Species) Accumulation on Leaves of P. persica

Drought can lead to oxidative stress and cell damage. As a result, under various treatment settings, we found cell death and ROS buildup in peach seedling leaves. Cell death and ROS accumulation in peach seedling leaves were compared using Evans Blue, DAB(3,3-diaminobenzidine), and NBT (Nitroblue tetrazolium) staining. Figure 4A,C,E show that under drought stress conditions, the leaves show a darker color and a larger staining area, suggesting that the cells in the leaves are damaged, the number of dead cells has increased, and the generation of ROS in the leaves has greatly increased. The stained area of leaves treated with exogenous LA was minimal under drought stress, showing that exogenous LA might mitigate drought damage. Stress can lead to the accumulation of reactive oxygen species and the intensification of membrane lipid peroxidation in plant leaves, and the content of MDA (malondialdehyde) in plant leaves’ cells can reflect the degree of oxidative damage. We examined the levels of O_2_^−^ and H_2_O_2_ to investigate the influence of LA pretreatment on oxidative alterations (Figure 4B,D). Compared with control plants, drought stress significantly increased O_2_^−^ and H_2_O_2_ contents in peach leaves by 45.33% and 32.47%, respectively. Compared with the drought-treated plants, the O_2_^−^ and H_2_O_2_ contents in peach leaves after LA application decreased by 32.71% and 28.42%, respectively. According to Figure 4F, drought stress significantly increased the MDA content in the leaves of peach seedlings. The MDA content in the leaves increased by 214.48% after drought treatment compared with CK. After exogenous LA application, the MDA content decreased by 140.82% compared with under the drought treatment.

### 2.5. Exogenous LA Enhanced the Antioxidant Capacity of P. persica under Drought Stress

Catalase (CAT), peroxidase (POD), superoxide dismutase (SOD), and ascorbate peroxidase (APX) are key antioxidant enzymes related to a plant’s restoration ability, which act by adjusting their activity to maintain the balance of active oxygen metabolism, thereby slowing the accumulation of active oxygen free radicals and membrane lipid peroxidation damage. As can be seen from Figure 5, compared with CK, the activities of CAT, POD, and SOD in peach leaves increased by 37.7%, 23.12%, and 13.69% after drought treatment, respectively. However, CAT activity decreased by 9.82%. Compared with the normal treatment, the activities of CAT, POD, SOD, and APX were increased by 10.58%, 24.35%, 13.56%, and 31.54%, respectively, in the LA-pretreated peach seedlings under drought stress. The results show that the exogenous application of LA increased the antioxidant enzyme activity of peach seedling leaves under drought stress, thereby removing reactive oxygen species and alleviating cell damage.

### 2.6. Sequencing of mRNA and Alignment to Reference Genome

We conducted transcriptional profiling on the leaves treated as above, using plants subjected to drought or control circumstances for seven days in order to better understand the impact of exogenous LA on the drought resistance of *P. persica*. Three duplicates of each cDNA library were created, as indicated in Table 1. Three biological copies of each sample yielded an average original reading of 40 to 47 million RNA sequences. We acquired an average of 39–46 million clean reads after removing low-quality reads, with more than 97% of these reads having Q30-grade sequences. More than 89% of them were unique mappings, compared to fewer than 2.8% that were multiple mappings. Each sample’s average GC content was almost 46%. Overall, the RNA sequence data quality justified further investigation.

### 2.7. Correlation of Gene Expression Level between Samples and Gene Expression Status

For each treatment of the three biological duplications, a correlation analysis was performed to determine the quality of the duplications (Pearson correlation coefficient was calculated using R language). Figure 6A shows that the three biological duplications of each treatment have strong correlation coefficients, which are all above 0.98, demonstrating that the three biological duplications are good and fulfill the statistical requirements. Principal component analysis (PCA) was performed using R software to assess sample repeatability and correlation (Figure 6B). The repeatability of the three biological replicates of each treatment is acceptable, and the differences between treatments are clear, indicating that the transcript results from this test are reliable. The standard has a |log2 (FPKM) ratio| ≥ 1, and the q value is ≤0.05. By comparing the gene expression intensity (FPKM) values between pairs of control (CK), drought stress treatment (CK-D), lauric acid treatment (LA), and lauric acid + drought stress treatment (LA-D), differentially expressed genes were identified. In CK-D vs. CK, LA-D vs. CK, LA-D vs. CK-D, LA-D vs. LA, LA-D vs. CK, and LA vs. CK-D, 5745, 4430, 1876, 5390, 3340, and 6936 DEGs (differentially expressed genes) were screened, including 2152, 1499, 1104, 1904, 2118, and 4135 up-regulated DEGs, while 2152, 2931, 772, 3486, 1222, and 2801 DEGs were down-regulated (Figure 6C,D).

### 2.8. GO Functional Analysis of DEGs

Functional enrichment analysis was carried out for DEGs in each group in order to determine the overall roles of these genes in various treatments. According to their GO annotations, the DEGs across the groups CK vs. CK-D, LA vs. CK, LA-D vs. CK-D, and LA vs. LA-D were split into three categories: biological process, cellular component, and molecular function.

The GO annotation results of DEGs between the CK and CK-D samples showed that 479 DEGs were annotated as biological processes (Figure 7A). In the biological process category, the minor categories with higher gene frequency and more significant GO annotation were carbohydrate metabolic process, response to oxidative stress, and photosynthesis. In total, 258 DEGs were annotated as cell components. Among the cell components, the minor categories with higher gene frequency and more significant GO annotations were extracellular region and cell wall. In total, 995 DEGs were annotated for molecular function. Among the molecular function categories, the minor categories with more significant gene frequencies than GO annotation were hydrolase activity, hydrolyzing O-glycosyl compounds, acting on glycosyl bonds, etc.

The GO annotation results show that 187 DEGs between CK and LA were annotated as biological processes (Figure 7B). Among the biological process categories, the minor categories with higher gene frequency and more significant GO annotation were defense response, response to biotic stimulus, and ion balance. In total, 117 DEGs were annotated as cell components. Among the major categories of cell components, the significant minor categories of GO annotation were extracellular region and cell periphery. In total, 487 DEGs were annotated for molecular function. Among the molecular function categories, the minor categories with higher gene frequency and more significant GO annotation were ADP binding, ion binding, etc.

The GO annotation results of differentially expressed genes between LA-D and CK-D show that 75 DEGs were annotated as biological processes (Figure 7C). Among the biological process categories, the minor categories with higher gene frequency and more significant GO annotation were defense response, cell wall organization or biogenesis, and response to oxidative stress. In total, 58 DEGs were annotated as cell components. Among the major categories of cell components, the significant minor categories of GO annotation were cell wall and external encapsulating structure. In total, 336 DEGs were annotated for molecular function. Among the molecular function categories, the minor categories with higher gene frequency and more significant GO annotation were ADP binding, heme binding, etc.

The GO annotation results of differentially expressed genes between LA and LA-D show that 535 DEGs were annotated as biological processes (Figure 7D). Among the biological process categories, the minor categories with higher gene frequency and more significant GO annotation were response to biotic stimulus, defense response, and response to stress. In total, 213 DEGs were annotated as cell components. Among the major categories of cell components, the significant minor categories of GO annotation were extracellular region and apoplast. In total, 779 DEGs were annotated for molecular function. Among the molecular function categories, the minor categories with higher gene frequency and more significant GO annotation were heme binding, tetrapyrrole binding, etc. 

### 2.9. KEGG Enrichment Analysis of DEGs

The biological processes carried out by various genes in vivo are coordinated. The pathways implicated in LA-mediated drought response in *P. persica* were further investigated using KEGG enrichment analysis, which was performed on the DEGs of each experimental group. The enrichment results are displayed in scatterplots (Figure 8).

The KEGG enrichment analysis results of DEGs in the CK vs. CK-D treatment group are shown in Figure 8A. As can be seen, the DEGs between the CK and CK-D treatments were mostly concentrated in the following areas: plant–pathogen interaction, plant hormone signal transduction, phenylpropanoid biosynthesis, starch and sucrose metabolism, amino sugar and nucleotide sugar metabolism, and cyanoamino acid metabolism.

The KEGG enrichment analysis of the DEGs between the LA and CK treatment groups is illustrated in Figure 8B. It can be seen that the DEGs were mainly concentrated in plant–pathogen interaction; phenylpropanoid biosynthesis; MAPK signaling pathway; glutathione metabolism; cyanoamino acid metabolism; and cutin, suberine, and wax biosynthesis.

The KEGG enrichment analysis of the DEGs between the LA-D and CK-D treatment groups is shown in Figure 8C. As shown on the map, the DEGs were mainly concentrated in plant–pathogen interaction, phenylpropanoid biosynthesis, MAPK signaling pathway, cyanoamino acid metabolism, and sesquiterpenoid and triterpenoid biosynthesis.

The KEGG enrichment analysis of the DEGs between the LA-D and LA treatment groups is shown in Figure 8D. As shown on the map, the DEGs were mainly concentrated in plant–pathogen interaction, phenylpropanoid biosynthesis, MAPK signaling pathway, cyanoamino acid metabolism, and sesquiterpenoid and triterpenoid biosynthesis. 

### 2.10. Hormone Signaling Pathway and Calcium (Ca^2+^) Signaling Pathway Gene Expression Analysis

It is generally understood that plant hormones play an important role in stress responses by coordinating many signaling pathways. Under drought stress, LA pretreatment impacted the transcription levels of genes linked to auxin, ABA, and salicylic acid signal transduction pathways, implying that LA may influence *P. persica*’s drought adaptation by interacting with different hormones. LA induced changes in many genes related to drought resistance in peach seedlings. A total of 76 hormone-related proteins and genes were found in the transcriptome data of differentially expressed genes, including 29 ABA-related proteins and genes (25 up-regulated and 4 down-regulated), 5 SA-responsive proteins and genes (4 up-regulated and 1 down-regulated), and 6 ABA-related proteins and transcription factors (3 up-regulated and 3 down-regulated) (Figure 9).

Notably, 21 up- or down-regulated DEGs were associated with the calcium (Ca^2+^) signaling pathway (Figure 10), indicating the integration of Ca^2+^ signaling mechanisms with LA-regulated responses to drought stress in peach seedlings. More specifically, there were seven, six and one DEGs belonging to the CNGC (cyclic nucleotide gated channel) family, CaM/CML (calmodulin/calmodulin-like protein) family, and CDPK (Ca^2+^ dependent protein kinase) family, respectively (Figure 10B). At the same time, a total of seven DEGS were changed in the FLS22 signaling pathway, including five flagellin receptors (FLS2), one promotor of mitogen-activated protein kinases (MEKK1), and one RBOH protein (RBOH).

## 3. Discussion

Plants face a variety of stress conditions in their growth and development stages, which will have different impacts and reduce crop yield [31,32]. Drought is the single most destructive environmental stress, and it causes more serious crop yield losses than any other stress [33]. Drought seriously affects the growth and development of plants, greatly reducing the crop growth rate and biomass accumulation [34]. The main consequences of drought are reductions in the rate of plant cell division and expansion, reduced leaf area, inhibitions in the rate of stem elongation and root proliferation, disrupted stomatal behavior, reductions in the nutrient relationship between water and crop yield, and reductions in water use efficiency [35,36,37]. When faced with drought stress, plants often adjust through a series of complex physiological and biochemical reactions [38].

Lauric acid (LA) is a kind of saturated fatty acid; the research on this compound mainly focuses on its antibacterial properties in relation to animals and plants [39,40]. It has also been found that LA exhibits certain biological activity, regulating nematode avoidance [41] and actively regulating the growth of soybean (*Glycine max* L.) [23]. By studying the changes in metabolic substances in loquat (*Eriobotrya japonica* Lindl.) under drought stress, it was found that LA accumulated significantly. It was speculated that LA might have similar functions to decanoic acid and abscisic acid (ABA) in loquat leaves, regulating the opening and closing of stomata to enhance drought resistance [28]. In this study, we treated *P. persica* subjected to drought stress with exogenous LA to investigate how LA alleviated the consequential damage. In addition, we identified the potential mitigation mechanism of exogenous LA in relation to drought stress through transcriptome analysis and the comparison of treatment groups. It has been reported that drought stress will also significantly reduce the growth of many important crops [42]. In this study, drought stress also significantly inhibited the fresh and dry weights and root activities of peach, as has been found in many studies of other plants, such as tomato [5], maize [35], willow [15], apple [43], and peach [30]. In this study, the fresh and dry weights and root activity following LA treatment were relatively higher than those following CK under drought stress, implying the effective alleviation of the adverse factors caused by drought.

The relative water content (RWC) of leaves may be the most appropriate measure of the physiological consequences of plant water status [44]. As shown in Figure 1, under drought stress, the RWC of LA-treated leaves was higher than that of untreated plants. This is consistent with the research results of [27], who reported that LA has the ability to maintain leaf water content under water deficit stress. Under normal conditions, the relative electrolyte leakage (REL) level of LA-treated plants did not change significantly compared with that of untreated plants. Under the condition of water shortage, the REL of LA-treated plants decreased by 15.32% compared with that of untreated plants (Figure 1). REL is an indicator of membrane damage. From our results, we can clearly see that LA plays an important role in preventing the cell membrane damage caused by water deficit stress [43].

Photosynthetic pigments are crucial for the absorption, transmission, and conversion of light energy during photosynthesis [17,43,45]. The content of photosynthetic pigment directly affects the photosynthesis of plants. Most studies have shown that the photosynthetic pigment content decreases with the increase in drought stress [46], but we found that LA treatment significantly inhibited this decline. Our results show that under certain drought stress conditions, applying the appropriate concentration of LA could help maintain higher chlorophyll and carotenoid contents in peach leaves. The increase in photosynthetic pigment content will ensure better light energy absorption and conversion efficiency in peach leaves. Photosynthesis is the basic function of plant growth, and it can be used to reflect the growth potential and drought resistance of plants [12,27,47]. Previous studies have shown that drought stress significantly reduces leaves’ photosynthetic rate, while LA treatment increases the photosynthetic rate [27]. We found that the net photosynthetic rate (Pn), stomatal conductance (Gs), intercellular carbon dioxide concentration (Ci), and transpiration rate (Tr) decreased to different degrees under drought stress. LA treatment can effectively alleviate drought damage, such that the Pn, Gs, Tr, and Ci of peach leaves are maintained at a higher level (Figure 2). These results indicate that LA could effectively maintain high levels of photosynthesis under certain degrees of drought stress.

Plants display corresponding adjustments and regulation mechanisms when under drought stress, which is of great significance in osmotic regulation [8,17,35]. Plants actively accumulate osmotic-regulating substances, increase cell fluid concentration, reduce cell water potential, and increase their water absorption capacity [48,49]. This study shows that the contents of osmotic adjustment substances of plants under drought stress, such as soluble sugar, soluble protein, proline, and free amino acids, increased to a certain extent, which indicates that peach seedlings can increase their ability to resist drought in this way (Figure 3). After applying LA at an appropriate concentration, this regulation mechanism was increased, thus slowing the damage caused by drought stress in peach seedlings.

When plants are subjected to stress, they produce a large amount of ROS, aggravate the degree of membrane lipid peroxidation, produce MDA, and destroy the integrity of cell membranes [50,51,52]. MDA content can reflect the degree of damage to cell membrane integrity [53]. When plants suffer from drought stress, the contents of H_2_O_2_, O_2_^−^, and MDA increase significantly [15,54]. In this study, after drought stress, these contents in the LA peach seedlings were slightly lower than in untreated plants, indicating that the protoplasm membrane system of LA-treated plants was less damaged and had better tolerance to drought (Figure 4). The leaves subjected to drought treatment were stained with cell death and reactive oxygen species chemicals, and the degree of staining was the greatest, indicating that the accumulation of H_2_O_2_ and O_2_^−^ in untreated plants was significantly higher than that in LA-treated plants, and the untreated plants suffered the most severe oxidative stress. Under drought stress, LA treatment can enhance the capacity for scavenging reactive oxygen species in peach leaf cells, and promote the reduction in H_2_O_2_ and O_2_^−^ accumulation, thus reducing cell damage or cell death, and enhancing plant stress resistance. The contents of ROS that accumulate in plants are low, the degree of oxidative damage to cells is light, and the integrity of cell membranes is good, which also enables the plant to maintain good photosynthesis [55]. Plants eliminate excessive active oxygen and reduce its toxicity by increasing the activity of their antioxidant enzymes. SOD, POD, CAT, and APX are important antioxidant enzymes in plants [18,19,34]. Under drought stress, the activities of SOD, POD, CAT, and APX in peach seedling leaves were significantly higher than those in the control plants (Figure 5). This indicates that plants can improve their drought resistance by increasing the activities of four of their antioxidant enzymes, eliminating excess ROS in their bodies, alleviating drought damage, and improving drought resistance [11,51]. After treatment with LA, the activities of four enzymes in the leaves of peach seedlings were further improved, indicating that LA can enhance the activities of antioxidant enzymes, thereby improving the antioxidant and anti-aging capacities of peach seedlings, and alleviating the toxic effects of drought stress. It was found that exogenous LA could reduce the oxidative damage of peach leaves and improve the activity of antioxidant enzymes to alleviate the damage caused by drought stress.

The secondary effects of drought stress are complex. The combination of RNA-Seq sequencing and DEG can be used to clarify the role of exogenous substances in directly or indirectly regulating plant stress resistance [32,43]. In order to better understand the mechanism of LA in drought-resistant plants, we used RNA-Seq technology to study the molecular mechanism of exogenous LA in relation to regulating peach drought resistance. This is the first study using transcriptome methods to explore LA and regulate plant stress resistance. Because the aims of this study include exploring the effects of exogenous LA on the molecular regulation mechanism of peach seedlings under drought stress, we focus on LA-D vs. CK-D. We identified 1876 DEGs between LA-D and CK-D: 1104 up-regulated and 772 down-regulated. GO analysis shows that most genes are involved in biological process categories, cell components, and molecular function.

As signaling molecules, plant endogenous hormones participate in the regulation of plant growth and development, play an important role in plant response to abiotic stress, and coordinate different signaling pathways [56,57]. In this study, the LA pretreatment affected the transcription levels of genes related to auxin, JA (salicylic acid), and ABA (abscisic acid) signal transduction pathways under drought stress, suggesting that LA may regulate the drought adjustment of peach seedlings through interactions with various hormones. After analysis, the majority of DEGs were allocated to the auxin pathway. Most of the genes related to auxin signal transduction were up-regulated, including 3 AUX1 (Auxin1) genes (2 up-regulated and 1 down-regulated), 12 AUX/IAA (Auxin/indole-3-acetic acid) genes (11 up-regulated and 1 down-regulated), 1 ABF (ABA response element binding factor) gene, 4 GH3 (Gretchen Hagen 3) genes (3 up-regulated and 1 down-regulated), and all 10 SAUR (small auxin-up RNA) genes. The critical role of auxin signaling genes in response to drought stress has also been demonstrated in other species. Auxin is a regulator of plant drought resistance, acting by regulating the AUX/IAA affinity of the auxin co-receptor, and it activates the transcriptional regulation of ARF-mediated early auxin response genes [58,59]. In addition to the ARF gene family, the auxin response genes include AUX/IAA, GH3, and SAUR. These three gene families are collectively referred to as auxin early response genes. Refs. [60,61] reported that the Aux/IAA protein OsIAA20, which is up-regulated under abiotic stress conditions, has a positive effect on the resistance to abiotic stress, enhances the growth of rice at different stages of development, reduces water loss, and increases pore closure. The authors of [62] reported the GH3 gene in three cotton varieties, and found it plays an important role in plants’ adjustment to drought and salt stress conditions. Ref. [63] showed that the overexpression of TaSAUR75 enhanced drought tolerance in Arabidopsis, and H2O2 accumulation in transgenic plants was lower than that in control plants.

SA, as a plant hormone, is an immune signal that causes systemic acquired resistance (SAR). The TGA (transcription factor TGA) and PR-1 (pathogenesis-related protein 1) gene families play an important role in the metabolism of plants in response to biological and abiotic stress [64,65,66]. It has been found that drought stress causes the up-regulation of all SlPR-1 genes, by up to 50 times, and our results show that the SlPR-1 gene played an active role in drought response. We found that under drought stress, LA treatment significantly up-regulates the related PR-1 gene.

The ABA signaling pathway is activated and regulated under drought stress and plays a key role in the regulation of plant resilience [67]. The four key genes of the ABA receptor, PYR/PYL, PP2C (protein phosphatases type 2C), SnRK2 (SNF1-related protein kinase 2), and downstream transport factor ABF (ABA-responsive element binding factors), are involved in the regulation of ABA signals [68]. They transmit ABA signals under the induction of ABA and stress, abscisic acid binds to PYR/PYL/ABA receptors, and the ABA receptor complex inhibits PP2C phosphatase activity, resulting in the PP2C-mediated inhibition of the release of SnRK2. Activated SnRK2 can phosphorylate downstream transcription factors and ion channels, and activate ABA signaling pathways and stress response processes, inducing various reactions such as increased ROS and pore closure [69,70]. In this study, LA caused changes in many genes associated with the ABA signaling pathway (Figure 9E,F).

The second messenger Ca^2+^ is widely involved in many types of signal transduction in plants [43]. Plants respond to many stimuli, such as the external environment and hormones, by starting gene expression through changes in the concentration of free Ca^2+^ in the cytoplasm, and ultimately improving stress adjustment [71,72,73]. In the current study, plant–pathogen interaction is one of the most significantly changed pathways (Figure 8C), and DEGs mapped to this pathway are mainly concentrated in Ca^2+^ pathway signals. Studies have shown that the Ca^2+^ signaling pathway is activated under drought stress, and this can improve the drought resistance of plants [57,74,75]. In addition, it has been found that osmotic stress induces an increase in exoplast Ca^2+^, which leads to the activation of CNGC (cyclic nucleotide-gated channel), thus causing responses in CaM/CML (Calmodulin/Calmodulin-like protein) and CDPK (calcium-dependent protein kinases) [43,57]. Increasing numbers of CaMs/CML family members have been identified and proven to be related to drought resistance [76]. The CGNCs in plants are a group of non-selective cation channels, which can promote the uptake of Ca^2+^ and thus mediate the acquisition of resistance [77]. Therefore, LA may improve the drought resistance of peach seedlings by activating the expressions of CNGC, CAM/CML, and CDPK family genes. In plants, oxidative stress has been proven to be related to CaM, which is a Ca^2+^-dependent activator [43]. Previous studies have found that SOD is a CaM-dependent enzyme [78]. Therefore, LA may increase the expression of the CaM/CML gene, activate the Ca^2+^ signaling pathway, and thus regulate antioxidant enzyme activity. The respiratory burst oxidase homolog (RBOH, also known as NADPH oxidase) is one of the key enzymes by which plants produce ROS. When cells are stimulated by external stress, they produce Ca^2+^, which is activated by the combination of RBOH’s EF structure [79]. Therefore, LA may regulate the activity of antioxidant enzymes through the regulation of the RBOH gene.

Flagellin-sensitive 2 (FLS2) is an important immunokinase receptor, which can effectively recognize the 22-amino acid peptide of plant–pathogen conserved peptide flg22 [80]. In recent years, important signal peptides and receptors that regulate plant response to drought have been discovered [81,82]. Based on our results, we propose that under LA treatment, FLS2 is involved in the perception of drought stress (Figure 10). Further verification of this potential function of FLS2 will provide new insights into the mechanism by which plants respond to drought stress.

In this study, using DEGs, multiple lncRNAs (long non-coding RNA) were predicted to explore the drought response induced by LA treatment. However, due to the lack of annotated information, it is difficult to determine the function of lncRNA in drought adjustment. However, the study of lncRNA in drought stress has become a hot topic [83,84]. We believe that the expression profile of lncRNA in response to drought is useful and valuable for the molecular breeding of peaches. At the same time, this study is the first to describe the response of lncRNA to drought in peaches under LA treatment.

## 4. Materials and Methods

### 4.1. Experimental Materials

Experiments were carried out at the experimental base of Shandong Agricultural University in 2022, located in Tai’an City, Shandong Province, China (36°17′7459″ N, 117°16′7712″ E). We selected uniform and plump peach seeds that had been steeped in a 400 ppm gibberellin solution for 24 h, before being put in seedling trays. We selected seedlings of comparable sizes that were free of disease and insect pests to plant in the basin when they reached approximately 5 cm in height. Each pot was square in shape (7 cm × 7 cm). The media in the pot comprised a 2:1 volume ratio blend of garden soil and vermiculite. Each pot of culture media weighed 200 g. The seedlings were managed on a regular basis.

### 4.2. Experimental Design

We selected 160 potted peach seedlings that had uniform development and were around 10 cm high. Among them, 80 pots of peach seedlings received 10 days of irrigation with 50 ppm LA (Shanghai macklin Biochemical Co., Ltd., Shanghai, China). Reagent concentrations were determined from the results of earlier preparatory studies. The results show that LA had no negative effect on the growth and development of peach seedlings. Then, the seedlings in each group were divided into two groups: conventional watering treatment (CK, LA) and natural drought treatment for 10 days (CK-D, LA-D), with 40 pots in each group. After 10 days of treatment, the leaves were carefully washed with tap water, frozen in liquid nitrogen, and stored at −80 °C for later use.

### 4.3. Determination of Plant Growth, RWC, and REL of Leaves

At the end of the drought treatment, peach seedlings were removed whole and cleaned, and fresh (FW) and dry (DW) weights were determined after oven-drying at 60 °C to a constant weight.

The phenyltetrazolium chloride (TTC) method was used for determination. The root tips were cut and weighed at 0.5 g. The sample was mixed with 0.4% TTC and pH 7.0 phosphate buffer in equal amounts up to 10 mL, sealed, and reacted in the dark at 37 °C for 4 h. The reaction was terminated by adding 1 mol∙L^−1^ sulfuric acid up to 2 mL. The sample was blotted and 4 mL of ethyl acetate and quartz sand was added, before the sample was ground well. The extract was transferred to a test tube and the residue was rinsed with ethyl acetate, and finally fixed with ethyl acetate up to 10 mL. Colorimetric assessment was undertaken at 485 nm. Three plants were chosen for each treatment, and two leaves were removed from each plant and promptly placed in an aluminum box of known weight, and the fresh weight (Wf) was determined. The leaves were submerged in distilled water for approximately 1 h before being removed and weighed to determine the saturated fresh weight (Wt) of the sample. The dry weight was then obtained by drying to constant weight (Wd). Relative water content (RWC) was determined using the following formula [84]: (Wf − Wd)/(Wt − Wd) × 100%.

The relative electrolyte leakage (REL) was measured with reference to [85], whose method was slightly modified. We punched 10 tiny discs from ripe leaves and placed them in a 20 mL centrifuge tube with 10 mL of deionized water. After 4 h at room temperature, we used a Raymag DDS-307 conductivity meter to test the solution’s conductivity, designated as S1; at the same time, we measured the conductivity of deionized water, denoted as S0. The centrifuge tube was then put in a 100 °C water bath for 20 min. After cooling, it was shaken well and we measured the conductivity S2. To represent the relative permeability of the plasma membrane, we calculated the relative electrolyte leakage using the following formula: REL (%) = (S1 − S0)/(S2 − S0) × 100%.

### 4.4. Determination of Photosynthetic Parameters and Leaf Photosynthetic Pigments

On the 14th day after drought stress, photosynthesis was monitored from 10:00 to 11:30 am. We measured the following parameters with a CIRAS-3 portable photosynthetic system (PP Systems, Massachusetts, USA): net photosynthetic rate (Pn), stomatal conductance (Gs), intercellular CO_2_ concentration (Ci), and transpiration rate (Tr).

Samples (0.2 g) were taken from fresh, clean peach leaves, and extracted for 24 h in 95% ethanol solution. We then used a Pharma-Spec UC-2450 ultraviolet spectrophotometer (Shimadzu, Kyoto, Japan) to measure the OD665, OD649, and OD470 of the extract. In order to calculate leaf chlorophyll and carotenoid contents, we used the methods of [43].

### 4.5. Determination of Osmotic Regulation Substances

The soluble sugar content was determined via the anthrone colorimetric method [47], and the absorbance at 620 nm was measured. The proline content was determined via the ninhydrin method [86], and the absorbance value of the toluene layer was measured using a spectrophotometer at 520 nm. Soluble protein content was determined via Coomassie brilliant blue staining [46]; we recorded the absorbance at 595 nm, and assessed the soluble protein concentration using the standard curve. The method for measuring the total amount of free amino acids was based on [87], with slight modifications, and we added 60% ethanol to 5 mL and tested the absorbance at 570 nm.

### 4.6. Histochemical Evaluation of Oxidative Damage and Cell Death

The peach leaves were stained with Evans Blue to measure cell death [55]. 3,3-diaminobenzidine (DAB) and nitroblue tetrazolium (NBT) histochemical staining was performed to observe the color development of superoxide anion and hydrogen peroxide, as described in the method of [88] with a slight modification.

### 4.7. Determination of Leaf Reactive Oxygen and Lipid Peroxidation

We measured the hydrogen peroxide contents (H_2_O_2_) of leaves using the method of [55]. Briefly, leaf samples (0.1 g) were placed in sterilized centrifuge tubes and then ground with liquid nitrogen. After centrifugation at 6000 rpm for 150 s, the samples were mixed with 1.5 mL of 0.1% trichloroacetic acid (TCA) and placed on ice. The samples were then centrifuged again at 12,000 rpm for 15 min at 4 °C. A 0.5 mL sample of supernatant was then collected and mixed with 0.5 mL of phosphate-buffered saline (PBS) and 1 mL potassium iodide (KI). The resulting solution was shaken well and held at 28 °C for 1 h, after which we measured the absorbance at 390 nm.

We measured the O_2_^−^ contents of leaves using the method of [55]. Briefly, we chopped 1 g samples of peach leaves and added 3 mL of phosphate buffer (pH = 7.8), placed them in an ice bath and ground them, and then centrifuged the samples at 4000× *g* for 15 min. The supernatant was collected and mixed with 0.1 mL of 10 mM hydroxylamine hydrochloride solution and incubated at 25 °C for 20 min. We then added 1 mL of 17 mM p-aminobenzene sulfonic acid and 1 mL of 7 mM a-naphthylamine solution and incubated the mixture at 25 °C for 20 min. After this, we added an equal volume of chloroform to extract the pigment and centrifuged the mixture at 10,000 rpm for 3 min. The pink extract was collected in order to measure the OD530.

Malondialdehyde (MDA), a biomarker of lipid peroxidation caused by oxidative stress, was measured using the method described by [46], with some modifications.

### 4.8. Determination of Leaf CAT, POD, SOD, and APX Activities

We weighed a mixed sample of 0.5 g, performed liquid nitrogen grinding, and added 4 mL of phosphate buffer at a concentration of 0.05 M pH 7.8 (0.3% EDTA with 0.1 mM, Triton-X100 and 4% of polyvinylpyrrolidone); the sample was then ground. Next, the sample was placed in a centrifugal tube, flushed twice with 6 mL buffer at 4 °C, then centrifuged at 12,000 rpm for 20 min; we then collected the supernatant and stored it at 4 °C until use.

Superoxide dismutase (SOD) activity was determined according to the method of [47], with some modifications. Peroxidase (POD) activity was determined according to the method of [55], with some modifications. Catalase (CAT) activity was determined according to the method of [47], with slight modifications.

### 4.9. RNA Preparation, Library Construction, and Sequencing

Total RNA was used as the input material for the RNA sample preparations. Briefly, mRNA was purified from total RNA using poly-T oligo-attached magnetic beads. Fragmentation was carried out using divalent cations at an elevated temperature in First Strand Synthesis Reaction Buffer (5X). First-strand cDNA was synthesized using random hexamer primer and M-MuLV Reverse Transcriptase (RNase H-). Second-strand cDNA synthesis was subsequently performed using DNA Polymerase I and RNase H. The remaining overhangs were converted into blunt ends via exonuclease/polymerase activities. After the adenylation of the 3′ ends of DNA fragments, adaptors with a hairpin loop structure were ligated to prepare the sample for hybridization. In order to preferentially select cDNA fragments of 370~420 bp in length, the library fragments were purified with the AMPure XP system (Beckman Coulter, Beverly, MA, USA). Then, PCR was performed with Phusion High-Fidelity DNA polymerase, Universal PCR primers and Index (X) Primer. Lastly, the PCR products were purified (AMPure XP system) and the library quality was assessed on the Agilent Bioanalyzer 2100 system.

### 4.10. Differential Expression Analysis

The differential expression analysis of four conditions/groups (three biological replicates per condition) was performed using the DESeq2 R package (1.20.0). DESeq2 provides statistical routines for determining differential expressions in digital gene expression data using a model based on the negative binomial distribution. The resulting *p*-values were adjusted using Benjamini and Hochberg’s approach for controlling the false discovery rate. Genes with an adjusted *p*-value ≤ 0.05 found by DESeq2 were assigned as differentially expressed. Prior to differential gene expression analysis, for each sequenced library, the read counts were adjusted using the edgeR program package through one scaling normalized factor. Differential expression analysis of two conditions was performed using the edgeR R package (3.22.5). The *p*-values were adjusted using the Benjamini and Hochberg method. A corrected *p*-value of 0.05 and absolute fold change of 2 were set as the threshold for significantly differential expression.

### 4.11. GO and KEGG Enrichment Analysis of Differentially Expressed Genes

The gene ontology (GO) enrichment analysis of differentially expressed genes was implemented using the clusterProfiler R package, in which gene length bias was corrected. GO terms with corrected *p*-values less than 0.05 were considered significantly enriched by differentially expressed genes. KEGG is a database resource used for understanding high-level functions and utilities of the biological system, such as the cell, the organism, and the ecosystem, via molecular-level information, usually through large-scale molecular datasets generated by genome sequencing and other high-throughput experimental technologies (http://www.genome.jp/kegg/ (accessed on 1 June 2022)). We used the clusterProfiler R package to test the statistical enrichment of differentially expressed genes in the KEGG pathway.

### 4.12. Statistical Analysis

SPSS 17.0 (IBM, New York, NY, USA) statistical analysis software was used to perform one-way ANOVA and the Duncan multiple comparison test. Statistical significance, designated as the lettered results in the charts below, was determined at 5% (*p* < 0.05). All chart data are shown as mean ± error bars’ deviation.

## 5. Conclusions

Through physiological and biochemical research and transcriptome data analysis, the effects of exogenous LA treatment on peach tree seedlings under drought conditions were studied. The results show the following: (i) LA treatment enhanced the growth of peach tree seedlings under drought stress and the relative water contents of leaves. (ii) The LA treatment of peach tree seedlings under drought stress improved the photosynthetic ability, chlorophyll content, and antioxidant enzyme activity, and reduced the H_2_O_2_, O_2_^−^ and MDA levels and conductivity, which may be related to the addition of LA. (iii) Compared with the untreated drought stress seedlings, a total of 1876 significant DEGs were identified, of which 1104 were up-regulated and 772 were down-regulated. In-depth analysis of the DEGs has shown that the LA-mediated drought response may involve the complex coordination of multiple plant hormone signals and synthesis pathways, as well as the activation of Ca^2+^ signals. At the same time, the lncRNAs involved in LA treatment and drought response were identified. This study provides a new method by which to reduce the impact of drought stress on plants and increase plant yield.

## Figures and Tables

**Figure 1 plants-12-01492-f001:**
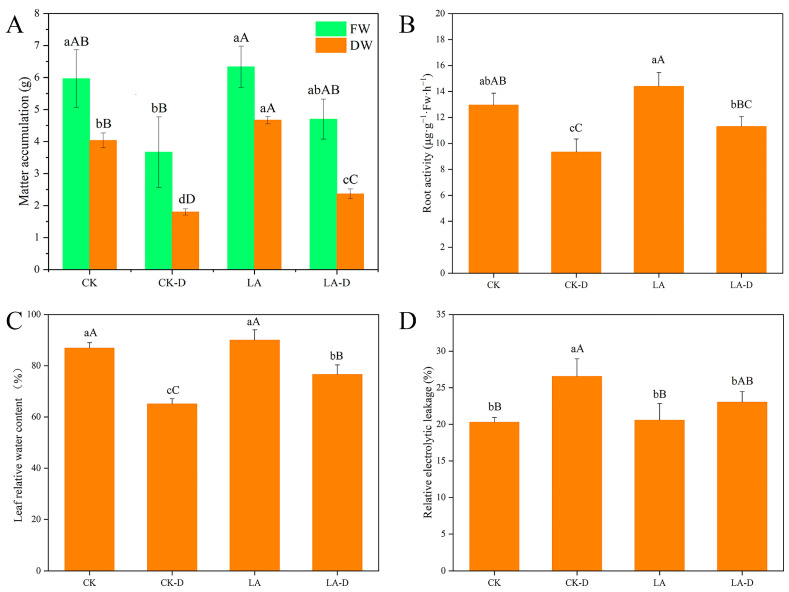
The effects of LA on *P. persica* growth under drought stress. Determination of dry and fresh weight of *P. persica* treated with LA under drought condition (**A**), determination of root activity (**B**). The effects of LA treatments on relative water content (**C**) and relative electrolyte leakage (**D**) in leaves under control and drought circumstances are shown in the graphs, while different small letter superscripts mean significant difference (*p* < 0.05), and different capital letter superscripts mean significant difference (*p* < 0.01). The same as below.

**Figure 2 plants-12-01492-f002:**
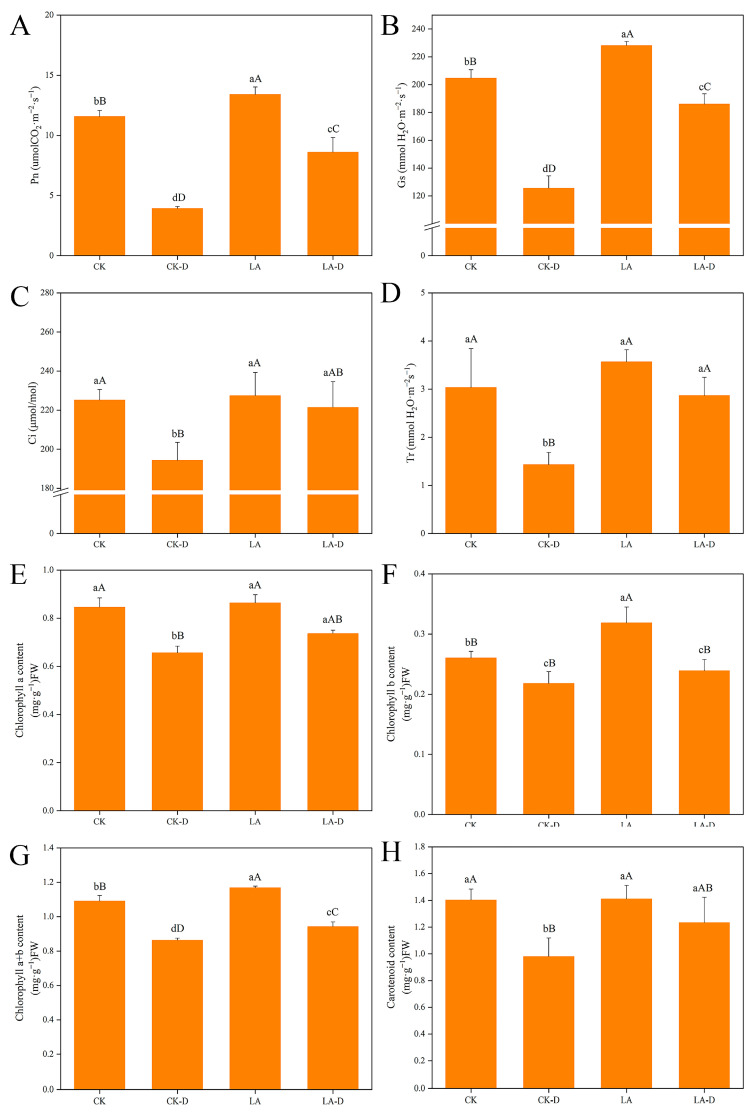
The effects of LA pretreatment on photosynthesis and chlorophyll content in drought-stressed *P. persica*. (**A**) net photosynthetic rate (Pn), (**B**) stomatal conductance (Gs), (**C**) intercellular carbon dioxide concentration (Ci), (**D**) transpiration rate (Tr), (**E**) chlorophyll a content (Chl a), (**F**) chlorophyll b content (Chl b), (**G**)total chlorophyll content (Chl a + b), (**H**) carotenoids (Car). Values represent means ± SD of three replicates. Different letters indicate significant differences according to Duncan’s multiple range tests.

**Figure 3 plants-12-01492-f003:**
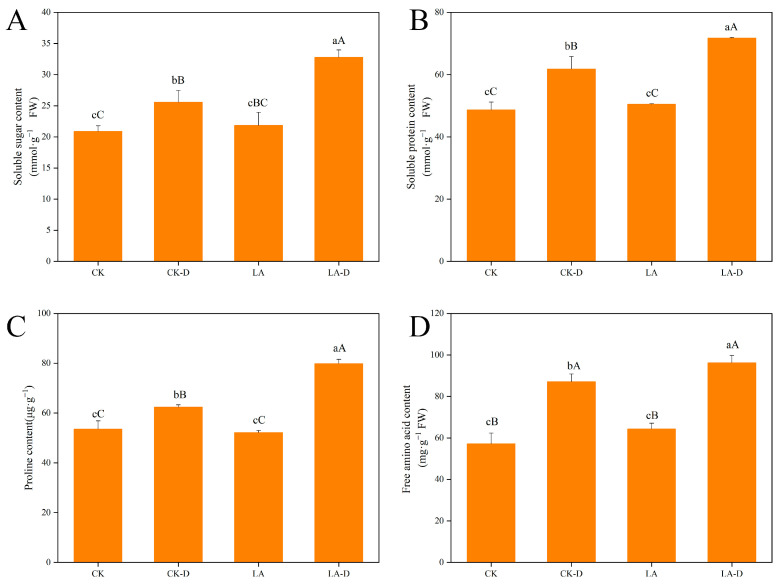
Effects of LA on osmotic regulation substances in *P. persica* under drought stress. (**A**) Soluble sugar content, (**B**) soluble protein, (**C**) proline content, (**D**) free amino acid content. Values represent means ± SD of three replicates. Different letters indicate significant differences according to Duncan’s multiple range tests.

**Figure 4 plants-12-01492-f004:**
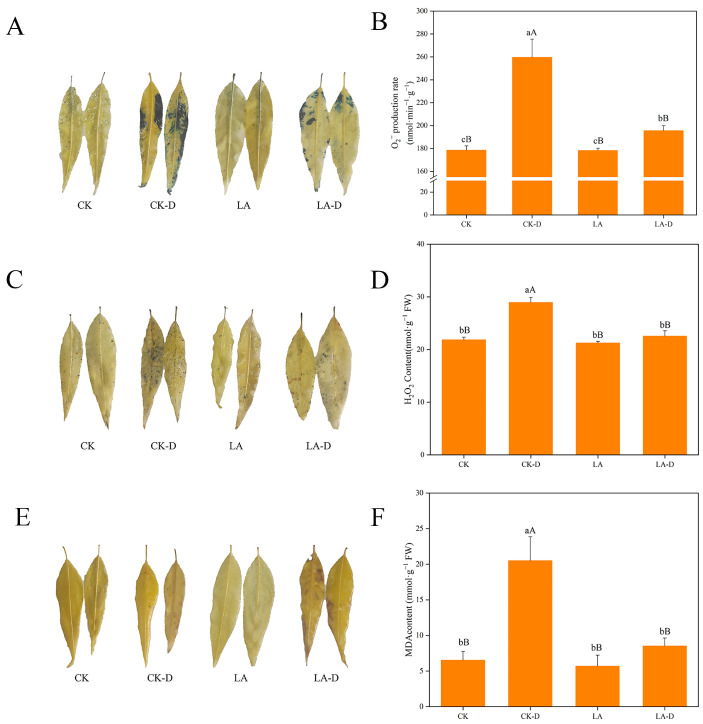
Effects of exogenous LA on cell death and ROS accumulation in leaves of *P. persica* under drought conditions. (**A**) Staining with Evans Blue, (**B**) effects of LA on O_2_^−^ contents in leaves under drought, (**C**) staining with nitroblue tetrazolium (NBT), (**D**) effects of LA on hydrogen peroxide (H_2_O_2_) content in leaves under drought, (**E**) staining with 3,3-diaminobenzidine (DAB), (**F**) MDA content. Values represent means ± SD of three replicates. Different letters indicate significant differences according to Duncan’s multiple range tests.

**Figure 5 plants-12-01492-f005:**
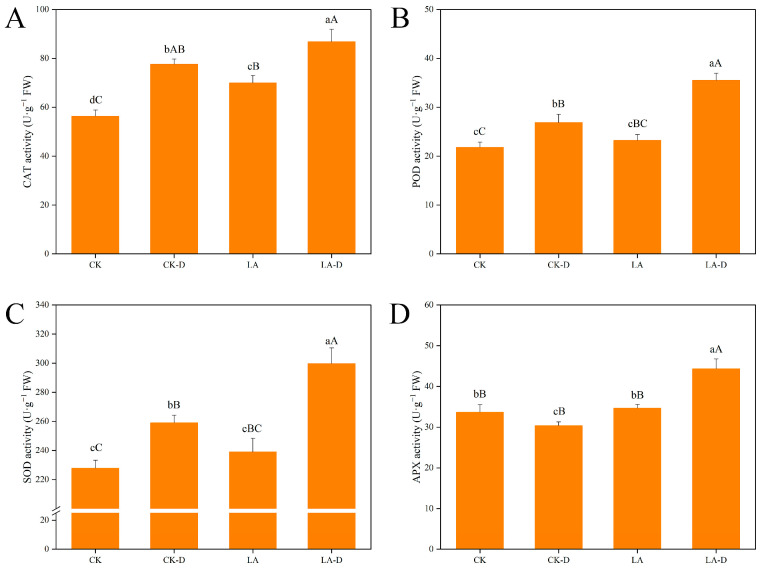
Effects of LA on antioxidant enzyme activity *P. persica* leaves under drought stress. (**A**) Catalase (CAT) activity, (**B**) peroxidase (POD) activity, (**C**) superoxide dismutase (SOD) activity, (**D**) ascorbate peroxidase (APX) activity. Values represent means ± SD of three replicates. Different letters indicate significant differences according to Duncan’s multiple range tests.

**Figure 6 plants-12-01492-f006:**
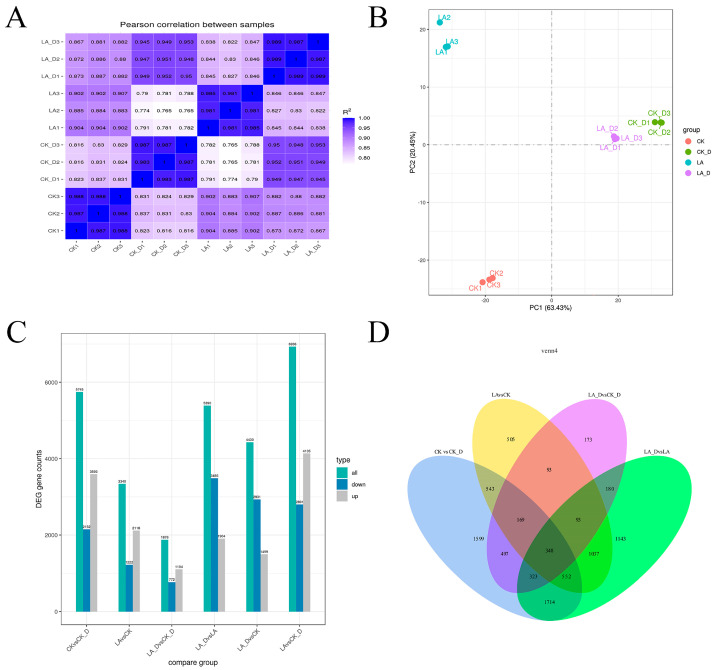
Correlation and differentially expressed genes among treatments at different sampling. (**A**) Heat map of the correlation coefficient between samples. (**B**) Principal component analysis (PCA) is also commonly used to assess inter-group differences and intra-group sample duplication. (**C**) Number of DEGs among the six comparison groups. (**D**) Venn diagram of differential gene expression analysis of DEGs.

**Figure 7 plants-12-01492-f007:**
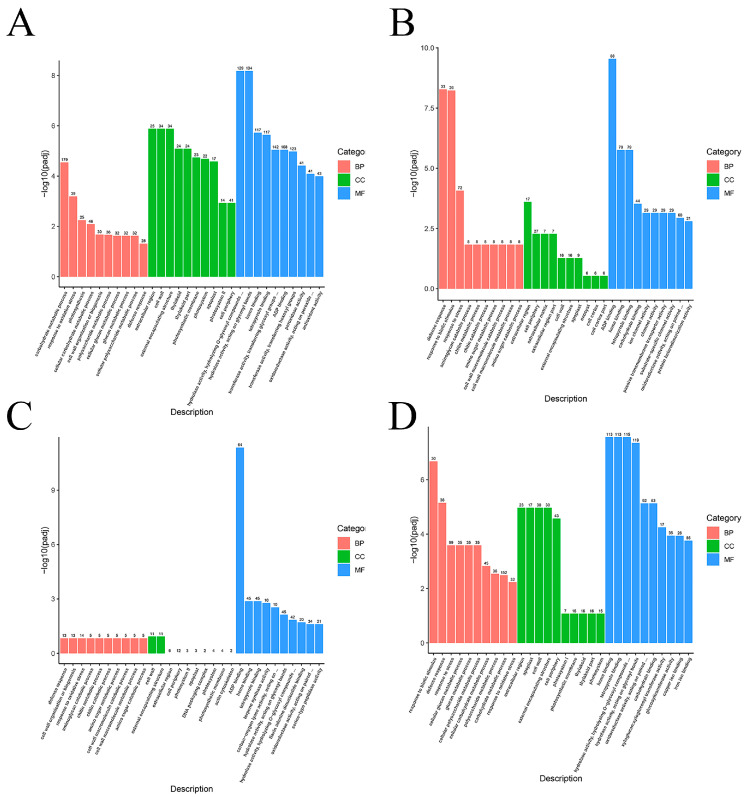
GO term enrichment analysis of DEGs in response to different stress treatments. (**A**) CK vs. CK-D; (**B**) LA vs. CK; (**C**) LA-D vs. CK-D; (**D**) LA-D vs. LA. The x-axis indicates the GO classifications, and the Y-axis indicates the number of genes in each classification. The top 30 GO terms with the most significant enrichment level are displayed in the order of *p*-value from small to large.

**Figure 8 plants-12-01492-f008:**
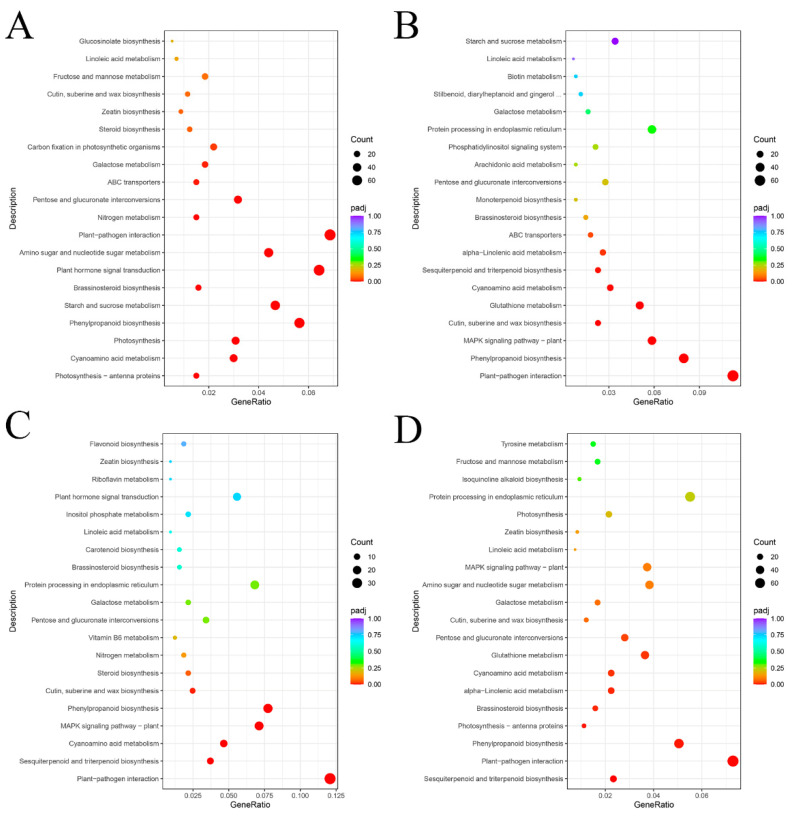
Kyoto Encyclopedia of Genes and Genomes (KEGG) pathway enrichment analysis of DEGs in response to different stress treatments. (**A**) CK vs. CK-D; (**B**) LA vs. CK; (**C**) LA-D vs. CK-D; (**D**) LA-D vs. LA. The X-axis shows the rich factor. Green represents a high q value, and red represents a low q value. The Y-axis shows the top 20 KEGG pathways. The bigger the size of the spot, the greater the number of DEGs enriched.

**Figure 9 plants-12-01492-f009:**
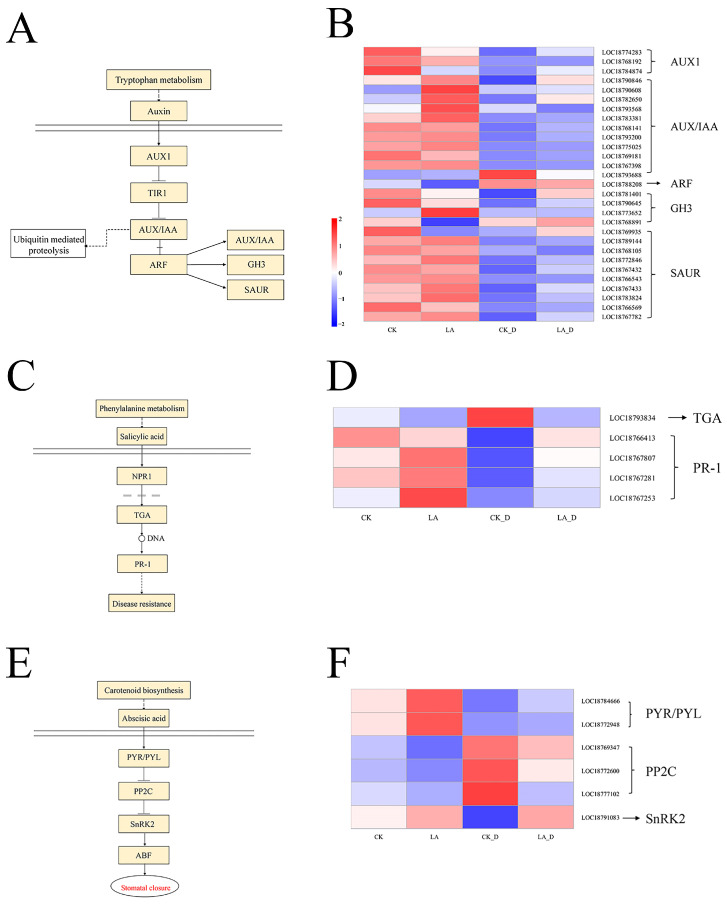
Effects of exogenous LA on hormone signaling in drought-stressed *P. persica*. Transduction of plant hormone signals in response to exogenous melatonin administration (**A**,**C**,**E**). Gene expression profiles for the signaling pathway for plant hormones (**B**,**D**,**F**). The expression levels were assessed by log2-transformed FPKM values.

**Figure 10 plants-12-01492-f010:**
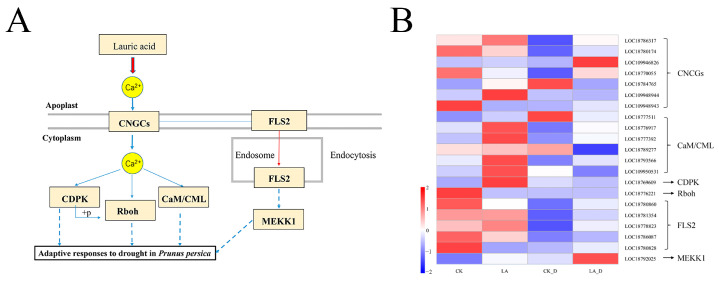
The Ca^2+^ signaling pathway induced by lauric acid in *P. persica* during drought stress (**A**) and a heat map of expression levels of CNGC, CAM/CML, CDPK, Rboh, FLS2, and MEKK1 (**B**). The expression levels were assessed by log2-transformed FPKM values.

**Table 1 plants-12-01492-t001:** Summary of RNA-Seq reads mapped to the peach genome.

Sample	Raw Reads	Total Reads	Unique Mapped	Multiple Mapped Reads	Q30 (%)	GC (%)
CK1	44,207,050	43,662,964	39,265,717 (89.93%)	1,162,481 (2.66%)	97.31	46.28
CK2	40,366,010	39,694,068	35,823,821 (90.25%)	1,078,507 (2.72%)	97.44	46.3
CK3	43,599,416	43,065,770	38,840,909 (90.19%)	1,131,967 (2.63%)	97.38	45.98
CK-D1	46,361,522	45,764,628	41,264,769 (90.17%)	1,158,338 (2.53%)	97.47	45.83
CK-D2	44,368,670	43,916,542	39,814,963 (90.66%)	1,093,090 (2.49%)	97.81	45.97
CK-D3	42,321,410	41,890,470	37,653,846 (89.89%)	1,073,290 (2.56%)	97.2	45.68
LA1	45,264,872	44,661,824	40,217,866 (90.05%)	1,225,178 (2.74%)	97.54	46.35
LA2	45,406,182	44,989,512	40,658,962 (90.37%)	1,144,441 (2.54%)	97.57	46.26
LA3	42,587,194	41,938,716	37,738,248 (89.98%)	1,135,371 (2.71%)	97.19	46.1
LA-D1	44,720,060	44,057,232	39,717,836 (90.15%)	1,150,437 (2.61%)	97.62	45.93
LA-D2	43,577,364	43,201,694	38,994,746 (90.26%)	1,077,302 (2.49%)	97.61	46.14
LA-D3	43,898,624	43,239,670	38,731,686 (89.57%)	1,136,746 (2.63%)	97.26	45.67

## Data Availability

The data presented in this study are available on request from the corresponding author.

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
