# Peer review of "Physiological and Transcriptomic Analyses of the Effects of Exogenous Lauric Acid on Drought Resistance in Peach (Prunus persica (L.) Batsch)"

_plants, 2023, doi:10.3390/plants12071492_

Round 1

Reviewer 1 Report

This English in this paper is very poor. It is extremely difficult to read and comprehend the meaning and thus I cannot judge the science. I recommend that the authors have the paper edited by someone for presentation and resubmit.

Author Response

Response to Reviewer 1 Comments

Thank you for your letter and the Reviewers’ comments concerning our manuscript

entitled “Physiological and transcriptomic analyses of the effects of ex-ogenouslauric acid on drought tolerance in peach (Prunus per-sica (L.) Batsch)”. (Manuscript ID: 2264422). Those comments are all valuable and very helpful for revising and improving our paper and have great guiding significance to our researches. We have studied the comments carefully and have made corresponding corrections which can meet with your approval hopefully. The main corrections in the paper and the responses to the Reviewer’s comments are as following:

Responses to the Reviewer’s comments:

Point 1: The manuscript also needs careful language correction.

Response 1: Thanks for your valuable question. Considering the Reviewer’s suggestion, We have carried out English polishing service and attached the certificate.

Reviewer 2 Report

The manuscript submitted by  Binbin Zhang et al. deals with the involvement of lauric acid in resistance to drought in peaches.

  The results obtained by the authors are interesting.  The problem is well stated and the purpose of the research is clearly explained and justified. The work is very complete, and the results obtained are good. The experiments are well-planned and performed. The results are well described in the manuscript. However, the manuscript must be improved before it can be accepted.

 Specific comments:

  1. Authors should use the term resistance (resilience)  instead of tolerance, in lines 3, 49, 46, 48, 85, 475, 513, 519

    Plants are resistant to drought because can tolerate dehydration and because can avoid dehydration. Be careful of using the term tolerance it is not the same as resistance.
  • The term adaptation/adopt (lines 363, 411, 467, 484, 509, 540 and Fig. 10A) should be changed to adjustment/adjust. Adaptation is an evolutionary adjustment. Adaptation includes developmental, morphological and physiological traits which help the growth under adverse conditions which usually refers to evolutionary created and genetically determined traits.
  • Abbreviations should be explained when the name is used for the first time, e.g. lauric acid line 98. Also, check other names.
  • Legends for Figures 1 and 4 are too long and should be rewritten

For example:

Figure 1. The effects of LA on P. persica growth under drought stress. Determination 6 of dry and fresh weight of P. persica treated with LA under drought condition(A), determination of root activity - How it was determined? ?? (B). The effects of LA treatments on relative water content(C) and relative electrolyte leakage(D) in leaves under control and drought circumstances are shown in the graphs, while with different small letter superscripts mean significant dif-110 ference(P<0.05), and with different capital letter superscripts mean significant differ ence(P<0.01). The same as below.

Figure 4. Effects of exogenous LA on cell death and ROS accumulation in leaves of P. persica under drought conditions. (A)Staining with Evans Blue, (B) Effects of LA on O2- contents in leaves under drought, (C) Staining with nitroblue tetrazolium (NBT), (D) Effects of LA on hydrogen peroxide (H2O2) content in leaves under drought, (E) Staining with 3,3-diaminobenzidine (DAB), (F)  MDA content. Values represent means ± SD of three replicates. Different letters indicate significant differences according to Duncan’s multiple range tests.

  • Lines 548-549 - …select …… peach seeds, select seedlings ….  – I think it should be used selected ….
  • Drought treatments  - this paragraph describe also LA treatments
  • What does it means natural drought treatment and conventional watering treatment (lines 560-561)? It should be carefully explained.

The manuscript also needs careful language correction.

Author Response

Response to Reviewer 2 Comments

Thank you for your letter and the Reviewers’ comments concerning our manuscript

entitled “Physiological and transcriptomic analyses of the effects of ex-ogenouslauric acid on drought tolerance in peach (Prunus per-sica (L.) Batsch)”. (Manuscript ID: 2264422). Those comments are all valuable and very helpful for revising and improving our paper and have great guiding significance to our researches. We have studied the comments carefully and have made corresponding corrections which can meet with your approval hopefully. The main corrections in the paper and the responses to the Reviewer’s comments are as following:

Responses to the Reviewer’s comments:

Point 1: Authors should use the term resistance (resilience)  instead of tolerance, in lines 3, 49, 46, 48, 85, 475, 513, 519. Plants are resistant to drought because can tolerate dehydration and because can avoid dehydration. Be careful of using the term tolerance it is not the same as resistance.

Response 1: Thanks for your valuable question. Considering the Reviewer’s suggestion, We have replaced these terms.( lines 3,51, 53, 55, 93, 522, 562, 570)

Point 2: The term adaptation/adopt (lines 363, 411, 467, 484, 509, 540 and Fig. 10A) should be changed to adjustment/adjust. Adaptation is an evolutionary adjustment. Adaptation includes developmental, morphological and physiological traits which help the growth under adverse conditions which usually refers to evolutionary created and genetically determined traits.

Response 2: Thanks for your valuable question. Considering the Reviewer’s suggestion, We have replaced these terms.( lines 400, 452, 513, 531, 558 and Fig. 10A)

Point 3: Abbreviations should be explained when the name is used for the first time, e.g. lauric acid line 98. Also, check other names.

Response 3: Thanks for your valuable question. Considering the Reviewer’s suggestion, We have checked and modified the abbreviations in the article.

Point 4: A Legends for Figures 1 and 4 are too long and should be rewritten. Figure 1. The effects of LA on P. persica growth under drought stress. Determination 6 of dry and fresh weight of P. persica treated with LA under drought condition(A), determination of root activity - How it was determined? ?? (B). The effects of LA treatments on relative water content(C) and relative electrolyte leakage(D) in leaves under control and drought circumstances are shown in the graphs, while with different small letter superscripts mean significant dif-110 ference(P<0.05), and with different capital letter superscripts mean significant differ ence(P<0.01). The same as below.

Figure 4. Effects of exogenous LA on cell death and ROS accumulation in leaves of P. persica under drought conditions. (A)Staining with Evans Blue, (B) Effects of LA on O2- contents in leaves under drought, (C) Staining with nitroblue tetrazolium (NBT), (D) Effects of LA on hydrogen peroxide (H2O2) content in leaves under drought, (E) Staining with 3,3-diaminobenzidine (DAB), (F)  MDA content. Values represent means ± SD of three replicates. Different letters indicate significant differences according to Duncan’s multiple range tests.

Response 4: Thanks for your valuable question. Considering the Reviewer’s suggestion, We made changes to the legend “Figure 1. The effects of LA on P. persica growth under drought stress. Determination of dry and fresh weight (A), root aitivity (B). The effects of LA treatments on relative water content(C) and relative electrolyte leakage(D) in leaves. Values represent means ± SD of three replicates. Different letters indicate significant differences according to Duncan’s multiple range tests.”

“Figure 4. Effects of exogenous LA on cell death and ROS accumulation in leaves of P. persica under drought conditions. (A)Staining with Evans Blue, (B) O2- contents, (C) Staining with NBT, (D) H2O2 content, (E) Staining with DAB, (F) MDA content. Values represent means ± SD of three replicates. Different letters indicate significant differences according to Duncan’s multiple range tests.”

We also supplemented the assay for root activity “Phenyltetrazolium chloride (TTC) method was used for the determination.The root tips were cut and weighed to 0.5 g. The sample was mixed with 0.4% TTC and pH 7.0 phosphate buffer in equal amounts for 10 mL, sealed and reacted in the dark at 37°C for 4 h. The reaction was terminated by adding 1 mol ∙ L-1 sulfuric acid for 2 mL. The sample was blotted out and 4 mL of ethyl acetate and quartz sand were added and ground well. The extract was transferred to a test tube and the residue was rinsed with ethyl acetate, and finally fixed with ethyl acetate to 10 mL. colorimetric at 485 nm.”(line 622-629)

Point 5: Lines 548-549 - …select …… peach seeds, select seedlings ….  – I think it should be used selected ….

Drought treatments  - this paragraph describe also LA treatments.

Response 5: Thanks for your valuable question. Considering the Reviewer’s suggestion, We have modified it to " Selected uniform and plump peach seeds that have been steeped in a 400ppm gibberel-lin solution for 24 hours before being put in seedling trays. Selected seedlings of com-parable size and free of illnesses and insect pests to plant in the basin when peach seedlings reach approximately 5 cm in height"(line600-606).

We have changed “4.2.Drought treatments” to “4.2. Experimental design” (line607)

Point 6: What does it means natural drought treatment and conventional watering treatment (lines 560-561)? It should be carefully explained.

Response 6: Thanks for your valuable question. Considering the Reviewer’s suggestion, We have modified it to” The results showed that LA had no negative effect on the growth and development of peach seedlings. Then the seedlings in each group were divided into two parts: con-ventional watering treatment (CK, LA) and natural drought treatment for 10 days (CK-D, LA-D), with 40 pots in each group. After 10 days of drought treatment, the leaves were carefully washed with tap water, frozen in liquid nitrogen, and stored at -80℃ for later use.”(line610-616)

Point 7: The manuscript also needs careful language correction.

回应7: 感谢您的宝贵问题。考虑到审稿人的建议,我们进行了英文抛光服务并附上了证书。

Round 2

Reviewer 1 Report

The writing of this paper is enormously improved, and it reads quite well. Communication is so important to convey great results. It is almost ready for publication with just a very few minor corrections.

“Figure” 1 is misspelled on line 206.

“activity” is misspelled on line 208.

2.2 This paragraph is continuous with the Figure 1 legend and needs to be separated.

Figure 10: what does “adjustment Ca++ signaling pathway” mean? (line 614)\

Line 619: remove “will” from the sentence.

Line 832: DGE should be DEG

Line 839: “which is a significantly differential gene”. . . doesn’t make sense.

Several places in the manuscript have P. persica not in italics.

Title 4.11 (Line 1277) should end with the word “genes”.

This is a really excellent study.

Author Response

Response to Reviewer 2 Comments

Thank you for your letter and the Reviewers’ comments concerning our manuscript

题为“前月桂酸对桃耐旱性影响的生理和转录组学分析”。(手稿编号:2264422)。这些意见都很有价值,对我们的论文的修改和改进很有帮助,对我们的研究具有重要的指导意义。我们仔细研究了这些意见,并进行了相应的更正,希望能得到您的批准。论文中的主要更正和对审稿人意见的回应如下:

对审稿人意见的回应:

第 1 点:“图”1 在第 206 行拼写错误。

响应 1: 感谢您的宝贵问题。考虑到审稿人的建议,我们修改了图 1 的图例。( 行324)

第 2 点:“活动”在第 208 行拼写错误。

响应 2: 感谢您的宝贵问题。考虑到审稿人的建议,我们修改了错误的拼写。( 326行)

要点3: 2.2 本段与图1图例连续,需要分开。

响应 3: 感谢您的宝贵问题。考虑到审稿人的建议,我们修改了段落格式。( 331行)

要点4: 图 10:“调整 Ca++ 信号通路”是什么意思?(第 614 行)\。

回应 4: 感谢您的宝贵问题。考虑到审稿人的建议,我们修改了段落格式“干旱胁迫期间月桂酸在P. per-sica中诱导的Ca2+信号通路”,以及CNGC,CAM / CML,CDPK,Rboh,FLS2和MEKK1(B)表达水平的热图。通过log2变换的FPKM值评估表达水平。( 896行)

要点5: 第619行:从句子中删除“将”。

回应5: 感谢您的宝贵问题。考虑到审稿人的建议,我们根据审稿人的要求进行了修改。(901行)

要点6: 第 832 行:DGE 应为度

回应6:感谢您的宝贵问题。考虑到审稿人的建议,我们根据审稿人的要求进行了修改。(1187行)

要点7: 第839行:“这是一个显着差异的基因”。没有意义。

回应7: 感谢您的宝贵问题。考虑到审稿人的建议,我们根据审稿人的要求进行了修改。(1194行)

要点8:手稿中的几个地方没有斜体。

回应8: 感谢您的宝贵问题。考虑到审稿人的建议,我们根据审稿人的要求进行了修改。(327,328,395,411,464,513,521,708,846,896,920行)

要点8:标题4.11(第1277行)应以“基因”一词结尾。

回应8: 感谢您的宝贵问题。考虑到审稿人的建议,我们根据审稿人的要求进行了修改。(1867行)
